# The Effect of Switchback Parameters on Root Pass Formation of Butt Welds with Variable Gap

**Hélio Antônio Lameira de Almeida [1], Felipe Ribeiro Teixeira [2]** ,
**Carlos Alberto Mendes da Mota [2] and Américo Scotti [3,4,]***

1   Center of Mechanics, Federal Institute of Education, Science and Technology of Pará, Belém-PA 66093-020, Brazil
2   Department of Mechanical Engineering, Federal University of Pará, Belém-PA 66075-110, Brazil
3   Laprosolda (Center for Research and Development of Welding Processes), Federal University of Uberlandia, Uberlândia-MG 38400-902, Brazil
4   Department of Engineering Science, University of West, S-461 32 Trollhättan, Sweden
*   Correspondence: americo.scotti@hv.se

**Abstract:** Root pass manufacturing in automated welding is still a challenge when the backing plate is not feasible. Using the concept of bead formation in an original way, the GMAW (Gas Metal Arc Welding) switchback technique was assessed against linear movement as a means of facing this challenge. Experimental work was applied, keeping the process parametrization and joint configuration, so that only the switchback parameters were modified, i.e., the stroke lengths and speeds. Thermography was used to estimate the effect of the switchback parameters on bead formation. The results showed the potential of the switchback technique as a means of favoring weld pool control. Surprisingly, the operational gap range is not necessarily larger when switchback is applied. The strong influence of stroke lengths and speeds on the process performance was characterized. In general, the results showed that linear movement leads to larger pools and deeper penetrations, more adequate for gaps with no clearances. Shorter stroke lengths and slower stroke speeds (intermediate pool size) better suit root gaps that are not too wide, while longer stroke lengths and faster stroke speeds (smaller pool size, more easily sustainable) are applicable to larger root gaps.

**Keywords:** GMAW; switchback; root pass; thermography

---

## 1. Introduction

The root pass consists of the first weld bead deposited in a groove. Deposition of a root pass requires a greater ability of the welder, since it must guarantee penetration, consistently and without perforations. The aspect and quality of a root pass are dependent of the forces that act directly on the weld pool. When there are no backing strips, the weld pool is supported by the shear stress induced by the surface tension gradient, which causes the liquid metal to flow from the center of the pool surface towards the joint edges. Considering the flat position, the weld pool is pressed down by the force induced by the plasma jet, the Lorentz force, and the gravity force. In this way, the welding current and the size of the weld pool tend to govern the stability of the root pass. However, conceptually, the formation of a root pass happens at two continuous yet distinct stages.

At the first stage, the arc is over the pool, heating and melting the metal (groove metal and filler). In the case of flat position welding, the arc jet and the gravitational and electromagnetic forces, the last one to a lesser extent, act pressing down the pool (although the arc jet avoids the pool collapse in the case of overhead position). The pool itself is sustained inside the groove by surface tension, which demands less free surface energy between the surrounding non-molten metal and the pool

than between the environment and the pool. Therefore, in the flat position, small size pools (lower gravitational force), a lower number of slags (which reduces the free surface energy between the environment and the pool), and low currents (less arc force pushing down the pool) favor the formation of a stable pool during the first stage.

It is worth noting two characteristics of this first stage. Although gravitational force can be assumed to depend only upon the molten metal mass, the action of the gravitational force is more pronounced in the overhead welding position, since there is a wider molten area on the lower surface of the pool than in the flat position. In the flat position, there is a more permanent backrest (joint faces) to the pool. Therefore, the positive effect of the arc jet towards sustaining the pool in the overhead welding position is counterbalanced by a higher negative effect of the gravitational force, and root pass in the overhead welding position is even more difficult. It is important to clarify that root faces with no gap demand a higher current to melt the metal at the groove root, also making more difficult (higher arc jet/electrical magnetic forces and larger molten metal volume) the formation of sound root passes. Too wide gaps, on the other hand, demand more molten metal to fill the volume, increasing the gravity force, yet requiring less current, since there is no need to melt the whole root face neighborhood.

A second characteristic is related to the mentioned reached pool stability at the end of the first stage. In fact, it would be more precise to say a pseudo-stable pool, since the molten metal is fluid and it would sag down eventually if the second stage of root pass formation would not take place. Regarding the second stage, as long as the arc is moving forward, the pool starts to cool down. Then, the molten metal viscosity increases, preventing the pool collapse before the full solidification (the reader must realize that viscosity is a dynamic property of a fluid, different from surface tension). Excessively fluid weld metal is not desirable for root pass in this concern, making the molten metal spillage faster (before full solidification). Therefore, small and deeply penetrated pools, as well as a reduced number of slags (all leading to faster solidification) favor the formation of sound root beads at the end of the second stage.

Considering all the above, the manual process with shielded metal arc welding (SMAW) is the most commonly used for root pass welding, especially with cellulose coating. The possibility of using transverse oscillation, relatively low currents and a short arc (low plasma jet) facilitates the control of the weld pool. Another widely used process for making a root pass is gas tungsten arc welding (GTAW), which, because of its independence between the material feed and the heat source, provides a skilled welder with the possibility of balancing the heat and the wire feeding, thereby controlling the weld pool. In these processes, the root pass can be made in different positions, but with relatively low welding speeds and deposition rates (slightly higher with SMAW) compared to other processes and welding techniques, generating limitations in terms of production. Finally, new versions of the gas metal arc welding (GMAW) process, using a controlled waveform format, brought new perspectives to build root passes, as demonstrated by Martikainen and Kah [1], in an interesting study which comparatively evaluated the quality and productivity of GMAW, SMAW, and GTAW root pass welding under different fit-up gaps. In a similar study, Adi et al. [2] demonstrated the ability to increase up to three times the travel speed of root pass with a version of a waveform-controlled process in comparison to SMAW or GTAW processes, with weld quality similar to GTAW process.

Nowadays, the reduction in the availability of skilled manual welders, an essential condition to build up root passes, has forced the use of automation. However, unlike manual welding, automation does not rely on the ability of the welder to control the weld pool. In order to get around this problem, one solution is to emulate the manual welder in automated operations, for tasks which are not so simple. Another solution would be to work on the concept of the root pass formation as summarized above. In this sense, the potentiality of the switchback technique with the GMAW process is presented.

The switchback technique consists of the periodic oscillation of the torch, a consequence of the arc (heat source), along the longitudinal direction of the groove during the welding operation. As illustrated in Figure 1, the arc cyclically moves forward with the weld pool along the joint by a certain linear amplitude, forming a first layer, so that it returns with a lesser length than that of the



forward stroke over the previous layer. Because the torch passes several times in the same location, the weld pool is heated slowly. Therefore, depending on the welding energy as well as the forward and backward strokes lengths and speeds, solid, liquid, and/or mushy phases will coexist in greater or lesser volumes. The angle of the torch in relation to the welding direction (with the arc pushing or pulling the pool) is another essential variable of the process.

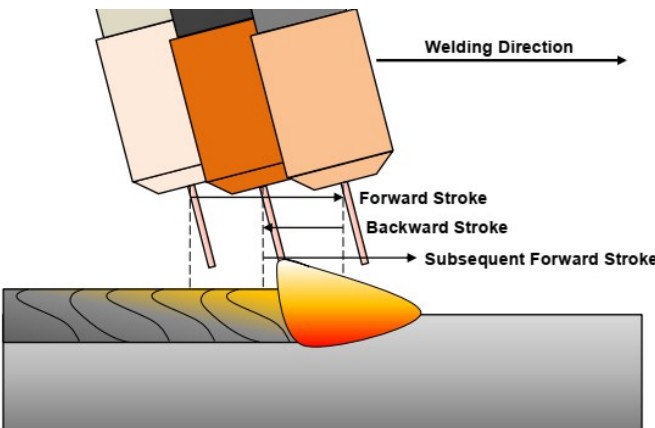

**Figure 1.** Periodic oscillation scheme of the torch with the switchback technique.

The potential of the switchback technique in terms of root pass formation was reported for the first time likely by Yamane et al. [3]. The authors obtained stable beads for large root gaps (between 2.3 mm and 4.9 mm and misalignments between 0.1 and 2.8 mm) in V-groove joints and without backing plates, through the continuous control of switchback technique parameters and GMAW process parameters. Based on this result, Yamane et al. [4] carried out another work and proposed the use of the switchback technique to control bead height and root pass, but this time with narrow gap welding. They claimed that a stable back bead is formed, since the arc pre-heats the root edges when moving forwards. When the welding torch is moved backwards, a suitable bead height is formed. In this way, the technique has been shown to be able to produce stable root passes also for narrow gap welding, provided that a suitable combination of switchback parameters is used. For this, numerical simulations were performed to obtain the adequate length of the course.

Further simulation studies on the switchback technique were realized by Kaneko et al. [5,6] in butt joints with thin plates. Heat conduction analysis was carried out to investigate the effect of the travel speed, as well as the backward stroke length of the torch on the weld pool. When the travel speed of the forward stroke was faster than that of the backward stroke, the weld pool length became shorter than a constant travel speed. On the other hand, if the backward stroke length became longer than half of the forward stroke length, continuous root passes could be obtained, regardless of such a disturbance as fluctuation of the arc length. Teixeira et al. [7] used experiments to study the switchback parametrization. The found operational envelope for the GMAW process with switchback was established with low values of speeds and lengths of forward and backward strokes.

In order to improve control of the root pass in welding without backing plates, Yamane et al. [8] designed and applied a cooperative control to integrate a robot, a welding power source, and a wire feed unit, so that the movements of the robot were adaptatively adjusted according to a root gap variation. Good quality welds were obtained by the control system. In other publications, Yamane et al. [9,10] gave more details of a switchback application in a V-groove joint with a root gap from 2 to 4 mm, without a backing plate. The assumption was that if the torch was rapidly moved forward, the heat source from the weld pool would disappear quickly and the temperature of the weld pool would fall, preventing burn-through. Therefore, in terms of the switchback technique, the proposed welding procedures were performed in two stages. In the first stage, the welding torch performed the forward and reverse strokes with high speeds, depositing the molten material only on the walls of the joint.

Thereafter, a new forward stroke was performed with the same prior length, however, with a reduced speed capable of merging and joining the previously deposited material on the walls of the joint. Thus, due to successfully combining the switchback technique with a transverse oscillation and a pulsed current GMAW, a good root pass could be obtained by setting an appropriate amplitude for the torch oscillation.

In order to improve control of the weld pool, Scotti et al. [11] registered a patent for a switchback welding device using the GMAW process. In this device, the weld pool control is performed by changing the operational mode of the process (polarity and/or metallic transfer) in synchrony with the position of the torch in the joint. However, unlike Yamane et al. [9,10], in which the torch oscillated transversely at the same time as the switchback movement, this device causes the torch to first travel one side of the groove face during the forward movement, changing to the other side in a backward movement, and concluding the cycle at the center of the groove with another forward movement. In this control device, the forward and backward lengths were the same. During displacement on the groove sides, the device makes the power supply impose a higher arc energy, while during the displacement over the center of the groove (root centerline), the control changes the operational mode to a lower energy. In this way, the device contributed to a greater heat distribution and control of the viscosity of the weld pool, avoiding the collapse of the same in different geometric tolerances in the root gap, either by misalignment or unevenness.

The possibility of increasing the limit travel speed in fillet welds (overspeed leads to a humping formation) has been also appointed as a beneficial characteristic of switchback. Aiming to use GMAW switchback to increase production with quality, Bonacorso et al. [12] compared linear conventional and switchback trajectories (forward speed four times faster than the backward and with pool pushing movement). However, the results showed a non-significant increase in the production with the switchback technique (limit increase of only 5%). This result contrasts with that mentioned in the review by Almeida et al. [13], where an increase of up to 60% in the limit travel speed was observed in the welding of a 3 mm thick steel overlap joint. Applying switchback to GTAW at high currents, Schwedersky et al. [14] showed that the incidence of humping-like discontinuities, typical of GTAW with high currents, was reduced. This improvement was clearer when the backward stroke length was more than half of the forward stroke length. Using thermography on the back of the plate, the authors suggested that the heat tends to penetrate into the plate less when switchback is applied, following the same trend observed in the simulations of Kaneko et al. [5,6].

As seen above, despite the fact that GMAW switchback has been acknowledged for some years, there is not much information in current literature about this technique and its application is unknown in the industrial environment. One reason for this fact would be the lack of robust information on parametrization, for example, about the reliability of the technique to manufacture root passes in butt joints with a variable root gap. Therefore, the objective of this work was to experimentally determine the potential gain in root gap tolerances (operational gap range) when switchback replaces the conventional linear progression welding (a typical application of GMAW), with the view of pool formation at the root pass.

## 2. Materials and Methods

### 2.1. Experimental Planning to Evaluate the Effect of Stroke Lengths and Speeds on the Operational Gap Range Limits

The methodology to reach the objective of this work was to comparatively weld butt welds (I joint) with a progressively increased root gap, using either switchback or the conventional linear movement techniques. During welding, switchback parameters were changed in order to optimize the weld pool control, as proposed and fundamentally explained in the introduction section. The criterion was to determine and compare the range size of the gap that each technique could perform without losing root pass quality, i.e., the root gap tolerance. In addition, a thermal analysis of each technique was implemented as a support to explain the results.

A robotic system moved the torch in both conditions, i.e., with switchback and with linear movements. All the welds were carried out by using the GMAW process, with the pulsed current at a low average value (145 A), to make root pass feasible. Table 1 presents the pulsed current parameters applied in this study. An AWS (American Welding Society) ER70S-6 wire (1.2 mm) was used as the filler metal. The arc was shielded by a gas blend of 95% Ar + 5% $O_2$, at a flow rate of 15 l/min. The angle of the torch in relation to the welding direction was 15° (pushing the pool), assuring a CTWD (contact tip to workpiece distance) of 18 mm, measured along the wire inclination. Therefore, during the switchback mode, the torch was pushing during the forward stroke and pulling during the backward stroke. The travel speed (TS) was kept constant at 4.2 mm/s. However, to establish comparisons with the conventional condition, the concept of equivalent travel speed ($TS_{eq}$), which is the combination of the forward and backward stroke speeds to deposit a bead with same length and time as the conventional condition, was applied to the switchback technique. The remaining parameters and variables were kept constant.

**Table 1.** Pulsed parametrization for the gas metal arc welding (GMAW) process.

| WFS (m/min) | $I_m$ (A) | $I_p$ (A) | $I_b$ (A) | $t_p$ (ms) | $t_b$ (ms) |
|---|---|---|---|---|---|
| 4.0 | 145 | 510 | 75 | 1.7 | 9.4 |

Note: WFS stands for wire feed speed; $I_m$ for mean current; $I_p$ for pulse current; $I_b$ for base current; $t_p$ for pulse time; $t_b$ for base time.

The test coupons were composed of two plain carbon steel (AISI 1020) bars, 140 mm × 50 mm × 3.3 mm thick (no machining on the bar lateral surfaces), as illustrated in Figure 2. Each bar was sand blasted and tack welded at the ends to get different gap ranges, nominally starting from an initial narrow root gap ($f_1$) and increasing to a final larger root gap ($f_2$). Due to the tack welds at the test coupon ends, the effective variation of the root gap happened in a test coupon length of 100 mm. If in that range of root gap a bead without irregularities was obtained, new test coupons would be welded from a gap slightly narrower (to verify repeatability) than the final root gap in which an accepted root was observed. For analysis of the test coupons, two parameters were evaluated: a minimum operational opening ($F_{min}$), which corresponded to the root gap value from which a total penetration was reached; and a maximum operational opening ($F_{max}$), which corresponded to the maximum root gap value achieved without any surface irregularities or burn throughs becoming apparent.

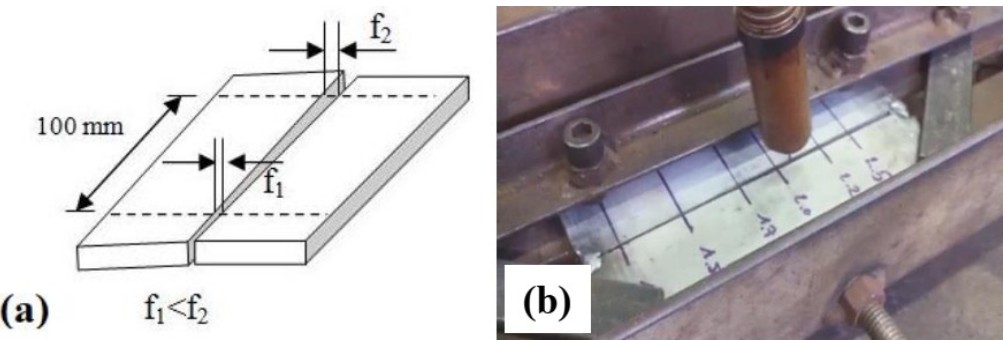

**Figure 2.** Test coupon showing the groove with an opening root gap: (**a**) schematic; and (**b**) in a fixture.

In order to evaluate the effect of the switchback technique on the weld pool control, an experimental plan was elaborated (Table 2). In this planning, the experiments were coded as follows: "L" for linear movement (no switchback) and "SB" for switchback movement. As indicated in Table 2, the linear condition was taken as the reference (conventional operational method). In this way, the forward and backward travel speeds ($TS_F$ and $TS_B$, respectively) were adjusted to make the equivalent travel speed of the switchback experiments with the same value as the linear movement. The experiment 1SB was

set with a longer forward stroke length (F) than backward stroke length (B), 30 mm and 20 mm (F = 1.5 B), respectively, yet keeping the same forward and backward stroke speeds ($TS_F = TS_B$). In the other experiments (2SB to 6SB), the forward and backward stroke lengths were kept constant, but with F twice as long as B, and the variables under study were $TS_F$ and $TS_B$. The variations in $TS_F$ and $TS_B$ were evaluated at three distinct modes, i.e., when the forward stroke speed was equal to the backward stroke speed (experiments 1SB and 2SB), when the forward stroke speed was less than the backward stroke speed (experiments 3SB and 4SB), and when the forward stroke speed was greater than the backward stroke speed (experiments 5SB and 6SB).

**Table 2.** Experimental planning (pulsed GMAW, 1.2 mm ER70S-6, CTWD of 18 mm, Ar + 5% $O_2$ and $I_m$ = 145 A).

| Exp. | TS (or $TS_{eq}$) (mm/s) | F (mm) | B (mm) | $TS_F$ (mm/s) | $TS_B$ (mm/s) |
|------|--------------------------|--------|--------|---------------|---------------|
| 1L   | 4.2  | –  | –  | –    | –    |
| 1SB  | 4.2  | 30 | 20 | 18.3 | 18.3 |
| 2SB  | 4.2  | 10 | 5  | 11.7 | 11.7 |
| 3SB  | 4.2  | 10 | 5  | 9.2  | 30.8 |
| 4SB  | 4.2  | 10 | 5  | 8.7  | 66.7 |
| 5SB  | 4.2  | 10 | 5  | 15.7 | 7.8  |
| 6SB  | 4.2  | 10 | 5  | 19.5 | 6.5  |

Note: TS stands for Travel Speed, F for forward stroke length, and B for backward stroke length.

## 2.2. Experimental Planning to Evaluate the Effect of Stroke Lengths and Speeds on the Size of the Weld Pool

In order to better understand the weld pool formation phenomenon when the switchback technique is present, video-thermography was applied. An FLIR (Forward Looking Infrared Radar) T440 camera was used, with an acquisition rate of 30 fps. The data compilation through the equipment software was carried out at the range of 200 °C to 1300 °C. The methodology was based on the comparison of thermal profile images and the maximum temperature from the back surface of the test coupons when the bead-on-plate depositions were welded, with or without switchback. With this approach, the effect of switchback parameters was qualitatively estimated by the size and shape of hottest area on the plate backside, assumed hereafter to be proportional (not the same) to the weld pool volume. This assumption is based on the fact that the arc energy was the same and test coupons had the same dimensions, consequently undergoing a similar heat flux. Furthermore, the maximum temperature was assessed as a means of quantifying the influence of the switchback parameters. It was also assumed that a larger pool volume would carry a higher heat content and that the distance from the pool bottom to the backside of the plate would turn shorter. Consequently, the temperature reached on the backside would also be higher.

The test coupons for this experimental phase were carbon steel plates (AISI 1020). The dimensions of the substrate were $200 \times 50 \times 4.6$ mm. The reason for not working with gaps and a thinner plate, as done in Section 2, was based on the objective of this work stage. The use of gaps and/or a thinner plate would make the heat flux into the plate to be predominantly 2D, whereas a thicker plate would make the temperature at the plate backside very low. Both cases would lead the experimental condition to be not sensitive enough to stress out the effect of the switchback parameters.

GMA welding was carried out over the surface of the test coupons (bead-on-plate) by using a pulsed current, at a travel speed of 4.2 mm/s. The torch was positioned at an angle of 15°, pushing the pool in the forward direction (and pulling during the backward stroke), keeping a CTWD (contact tip to work distance) of 18 mm, measured along the wire (1.2 mm AWS ER70S-6) inclination. The arc was shielded by a gas blend of 98% Ar + 2% $O_2$, at a flow rate of 15 l/min. Table 3 presents the pulsed current parameters applied in this study.

**Table 3.** Pulsed parametrization for the GMAW process.

| WFS (m/min) | $I_m$ (A) | $I_p$ (A) | $I_b$ (A) | $t_p$ (ms) | $t_b$ (ms) |
|:---:|:---:|:---:|:---:|:---:|:---:|
| 4.0 | 145 | 400 | 50 | 3 | 8 |

Note: TS stands for travel speed, F for forward stroke length, and B for backward stroke length.

Only two typical switchback conditions were selected for the thermographic assessment, both with $TS_F > TS_B$, as presented in Table 4. The switchback comparison was established between the long stroke lengths (T1SB) and short stroke lengths (T2SB). A condition with linear movement (no switchback) was also used as a reference (T1L). In addition, similar to Section 2, the travel and equivalent speeds took the same value, that is, 4.2 mm/s.

**Table 4.** Experimental planning (pulsed GMAW, 1.2 mm ER70S-6, CTWD of 18 mm, Ar + 5% $O_2$ and $I_m$ = 145 A).

| Exp. | TS (or $TS_{eq}$) (mm/s) | F (mm) | B (mm) | $TS_F$ (mm/s) | $TS_B$ (mm/s) |
|:---:|:---:|:---:|:---:|:---:|:---:|
| T1L | 4.2 | – | – | – | – |
| T1SB | 4.2 | 30 | 20 | 24.2 | 15.8 |
| T2SB | 4.2 | 10 | 5 | 16.2 | 7.8 |

Note: WFS stands for wire feed speed; $I_m$ for mean current; $I_p$ for pulse current; $I_b$ for base current; $t_p$ for pulse time; $t_b$ for base time.

## 3. Results and Discussion

### 3.1. Effect of Stroke Lengths and Speeds on the Operational Gap Range Limits

Figure 3 presents the welded joints for the experiment without switchback technique (1L). In this condition, the root gaps varied from 0 to 2.5 mm, reached accumulatively after using three different test coupons. Based on this figure, full penetration happened from a minimum operational opening ($F_{min}$) of 0.3 mm and remained stable up to a maximum operational opening ($F_{max}$) of 1.7 mm. Therefore, the root gap tolerance was 1.4 mm (from 0.3 mm to 1.7 mm) for the process with linear movement.

| Test cupons | $f_1$ - $f_2$ (mm) | $F_{min}$ (mm) | $F_{max}$ (mm) | Aspects of the plate face and back |
|:---:|:---:|:---:|:---:|:---:|
| J1 | 0 – 1.7 | 0.3 | 1.7 | |
| J2 | 0.5 – 2.0 | 0.5 | 1.7 | |
| J3 | 1.0 – 2.5 | 1.0 | 1.7 | |

**Figure 3.** Quantification of the operational gap range for the experiment 1L (linear torch movement).

Figures 4 and 5 present the welded joints for the switchback experiments with forward stroke speed equal to the backward stroke speed ($TS_F = TS_B$). It is important to note that the bead appearance with switchback resembles a string of beads with overlap parts, as stated by Kaneko et al. [5]. For experiment 1SB (Figure 4), which presented longer forward and backward stroke lengths, full penetration happened from a minimum operational opening ($F_{min}$) of 1.7 mm and remained stable up to a maximum operational opening ($F_{max}$) of 3.0 mm. For root gaps narrower than 1.7 mm, there was a lack of penetration, while for values above 3.0 mm, there was perforation. Thus, the condition with switchback 1SB provided a root gap tolerance of 1.3 mm (from 1.7 mm to 3.0 mm), which is approximately the same as the experiment 1L (1.4 mm). However, the condition with linear movement (1L) was operational for narrower gaps (from 0.3 mm to 1.7 mm), while the condition with switchback movement and longer strokes (1SB) showed to be operational for wider gaps (from 1.7 mm to 3.0 mm). It is understood in this switchback condition the first stroke is colder than when linear movement is carried out, contributing to the stability of the root pass in wider gaps, where penetration is facilitated.

| Test cupons | $f_1 - f_2$ (mm) | $F_{min}$ (mm) | $F_{max}$ (mm) | Aspects of the plate face and back |
|---|---|---|---|---|
| J4 | 1.7 – 3.2 | 1.7 | 3.2 |  |
| J5 | 2.2 – 3.7 | 2.2 | 3.0 |  |

**Figure 4.** Quantification of the operational gap range for the experiment 1SB (switchback torch movement).

In the case of experiment 2SB (Figure 5), when the forward and backward stroke lengths were significantly reduced and the condition $TS_F = TS_B$ was maintained, full penetration happened from a root gap of 1.0 mm to 3.0 mm. Contrasting to experiment 1SB, there was also penetration for narrower gaps (<2.0 mm). Smaller stroke lengths resulted in a weld pool that was less elongated, and, therefore, hotter than condition 1SB, justifying the observed behavior. In addition, as well as experiment 1SB, experiment 2SB had a greater maximum operational opening than experiment 1L. However, it is worth noting that this condition presented a lack of penetration in some isolated points with root gaps less than 2.0 mm. Therefore, from a robustness point of view, $F_{min}$ was assumed for experiment 2SB to be 2.0 mm, with a root gap tolerance of 1.0 mm and $F_{max}$ of 3.0 mm.

The results up to now suggest that the switchback movement influences the quality of the root pass and the parameters of the technique play an important role in this phenomenon. This is the reason for planning the second and third experimental modes, i.e., with $TS_F < TS_B$ (experiments 3SB and 4SB) and with $TS_F > TS_B$ (experiments 5SB and 6SB). Figure 6 shows the welded joints from experiments 3SB and 4SB. The minimum and maximum operational openings were very similar ($F_{min}$ = 1.1 mm and $F_{max}$ = 2.5 mm), although the weld root was more regular for 4SB. Thus, the root gap tolerance was assumed to have reached 1.4 mm when the condition $TS_F < TS_B$ was used. This value was approximately the same as that achieved by experiments 1L and 1SB (1.4 mm and 1.3 mm, respectively) and greater than that obtained with experiment 2SB (which was 1.0 mm). This means

that, within the parameters evaluated, the conditions with $TS_F < TS_B$ did not improve the performance of the technique in relation to the formation of the root pass.

| Test cupons | $f_1$ - $f_2$ (mm) | $F_{min}$ (mm) | $F_{max}$ (mm) | Aspects of the plate face and back |
|---|---|---|---|---|
| J6 | $0 - 1.7$ | 1.0 | 1.7 | |
| J7 | $0.5 - 2.0$ | 1.0 | 2.0 | |
| J8 | $1.0 - 2.5$ | 1.2 | 2.5 | |
| J9 | $1.5 - 3.0$ | 2.0 | 3.0 | |

**Figure 5.** Quantification of the operational gap range for the experiment 2SB (switchback torch movement).

| Test cupons | $F_{min}$ (mm) | $F_{max}$ (mm) | Speeds (mm/s) | | Aspects of the plate face and back |
|---|---|---|---|---|---|
| | | | $TS_F$ | $TS_B$ | |
| J10 (3SB) | 1.1 | 2.5 | 9.2 | 30.8 | |
| | | | \($TS_F$ =0.3 $TS_B$\) | | |
| J11 (4SB) | 1.0 | 2.5 | 8.7 | 66.7 | |
| | | | \($TS_F$ =0.13 $TS_B$\) | | |

**Figure 6.** Quantification of the operational gap range for experiments 3SB and 4SB (switchback torch movement), where $TS_F < TS_B$.

Figure 7, in turn, shows the experiments in which forward stroke speeds were faster than the backward stroke speeds (experiments 5SB and 6SB). For this setting, it was observed that $F_{min} = 1.7$ mm and $F_{max} = 2.5$ mm, resulting in a root gap tolerance of 0.8 mm. This tolerance is shorter than that

obtained for the experiments with $TS_F < TS_B$ (3SB and 4SB), with $TS_F = TS_B$ (1SB and 2SB), and with linear movement (1L). In this way, within the parameters evaluated, the conditions with $TS_F > TS_B$ did not improve the performance of the technique in relation to the formation of the root pass.

| Test cupons | $F_{min}$ (mm) | $F_{max}$ (mm) | Speeds (mm/s) | | Aspects of the plate face and back |
|---|---|---|---|---|---|
| | | | $TS_F$ | $TS_B$ | |
| J12 (5SB) | 1.7 | 2.5 | 15.7 | 7.8 | |
| | | | $(TS_F = 2\ TS_B)$ | | |
| J13 (6SB) | 1.7 | 2.5 | 19.5 | 6.5 | |
| | | | $(TS_F = 3\ TS_B)$ | | |

**Figure 7.** Quantification of the operational gap range for experiments 5SB and 6SB (switchback torch movement), where $TS_F > TS_B$.

Table 5 displays a summary of the root gap ranges with full penetration for each tested condition. As seen in the mentioned table, in terms of root gap tolerance for the switchback conditions, the welded joints with a forward stroke speed lower than the backward stroke speed (3SB and 4SB) were those that presented root passes which were more stable, yet at the same level as one of the conditions with forward stroke speed equal to the backward stroke speed (1SB) and the condition with without application of the switchback (1L). On the other hand, based on results from elsewhere, improvements could be made through slight adjustments of the switchback parameters. According to the simulations carried out by Kaneko et al. [5,6], also in thin plate butt welding, and the results obtained by Schwedersky et al. [14], using GTAW at high currents, if the backward stroke length became longer than half of the forward stroke length (B > F/2), continuous root passes could be obtained. In the present cases, the only condition that fulfilled this setting (B > F/2) was the 1SB experiment. If 1SB is compared with 2SB (also with $TS_F = TS_B$, yet with B = F/2), there is an operational gap enlargement.

**Table 5.** Summary of the outputs of the experiments, as planned in Table 2.

| Exp. | Switchback Conditions | | | $F_{min}$ (mm) | $F_{max}$ (mm) | Operational Gap Range (mm) |
|---|---|---|---|---|---|---|
| 1L | – | – | – | 0.3 | 1.7 | 1.4 |
| 1SB | $TS_F = TS_B$ | F = 30 | B = 20 | 1.7 | 3.0 | 1.3 |
| 2SB | $TS_F = TS_B$ | F = 10 | B = 5 | 2.0 | 3.0 | 1.0 |
| 3SB | $TS_F < TS_B$ | F = 10 | B = 5 | 1.1 | 2.5 | 1.4 |
| 4SB | $TS_F < TS_B$ | F = 10 | B = 5 | 1.0 | 2.5 | 1.5 |
| 5SB | $TS_F > TS_B$ | F = 10 | B = 5 | 1.7 | 2.5 | 0.8 |
| 6SB | $TS_F > TS_B$ | F = 10 | B = 5 | 1.7 | 2.5 | 0.8 |

Note: Pulsed GMAW (AWS ER70S-6, Ar + 5% $O_2$ and CTWD of 18 mm, $I_m$ = 145 A and $TS_{eq}$ = 4.2 mm/s); F for forward stroke length (mm), B for backward stroke length (mm), $F_{min}$ for the root gap value from which a total penetration was reached, and $F_{max}$ for the maximum root gap value achieved without any surface irregularities or burn throughs becoming apparent.

Still based on Table 5, comparing switchback (SB) and linear movement (1L) experiments, there is a considerable difference between the welded joints. The switchback, for same arc energy, is not able to reach full penetration when the root gap is too narrow. The reason for this behavior would be



related to the high forward and backward stroke speeds used in relation to the travel speed of the linear movement experiment. A lower heat input in the first stroke (forward) decreases the digging power of the weld pool in the substrate and, consequently, the penetration. According to Kaneko et al. [5,6], this phenomenon also affects the shape of the pool—mainly the length. It is important to mention that in the present work, similar welding conditions to those used by Kaneko et al. [5] ($I_m$, $TS_{eq}$, Wire Feed Speed (WFS), stroke lengths and plate material) were applied, the differences being that the plate was slightly thicker and there were lower peak and base currents, as well as shorter peak and base durations. However, Kaneko et al. [5] stated that a higher root stability was observed for the condition of forward speeds faster than backward speeds ($TS_F > TS_B$). These conditions are more similar to those of 5SB and 6SB, for which the operational gap ranges were the smallest. Nevertheless, Kaneko et al. [5] did not determine the operational gap range, which suggests that the relationship between $TS_F$ and $TS_B$ is not the governing parameter, at least not alone (the stroke lengths F and B must also be considered).

### 3.2. Effect of Stroke Lengths and Speeds on the Size of the Weld Pool

Figure 8 illustrates the superficial aspects of the three bead-on-plate depositions performed at this stage of the work. The superficial aspect of the beads using switchback are uniformly scaled, in contrast to the bead using linear movement, which presents a smooth aspect (typical of pulsed GMAW). However, they are regular along the whole length, with the space between each scale proportional to the stroke lengths (T1SB is more spaced than T2SB). These characteristics support the assumption that stable weldments were performed using the three conditions of Table 4.

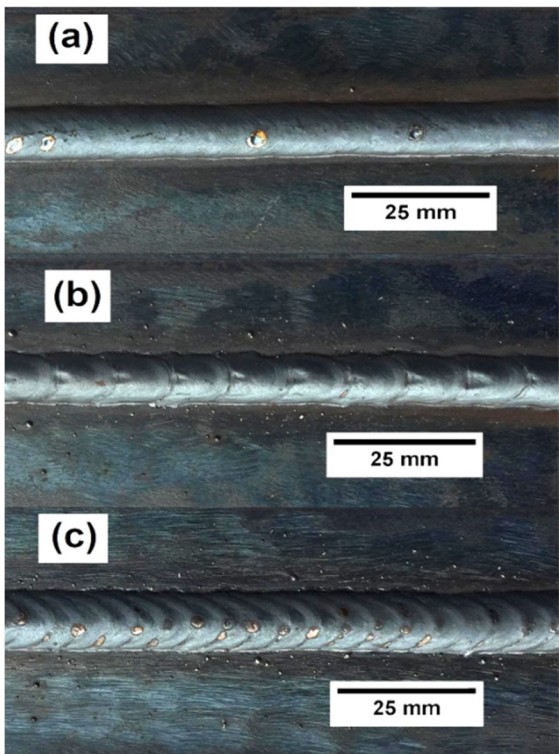

**Figure 8.** Superficial aspects of the beads deposited on plates (same WFS and TS) for thermographic assessment: (**a**) T1L (linear); (**b**) T1SB (switchback with F = 30 mm and B = 20 mm); (**c**) T2SB (switchback with F = 10 mm and B = 5 mm).

Figures 9–11, in turn, indicate the thermal profile on the backside of the plates for each deposition (T1L, T1SB, and T2SB, respectively) at three different welding positions (beginning, middle, and end of the bead), emphasizing the temporal progression of the welding torch. The hottest areas (rich red) were around 800 °C, surrounded by decrescent isotherms up to 200 °C (magenta). In general, the red and

yellow areas of each condition remained the same size regardless the heat source position, showing that the weld reached the steady state from the beginning. However, the farther the heat source was from the welding starting point, the more elongated and enlarged were the isotherms related to lower temperatures (from heavy green towards light cyan, bright blue, and magenta), showing, as expected, a longer time to converge to an equilibrium final temperature of the plate. As explained in the first paragraph of Section 3.1, the rich red isotherm was proportional to the weld pool size. Thereafter, the thermal profile analysis was concentrated in this area.

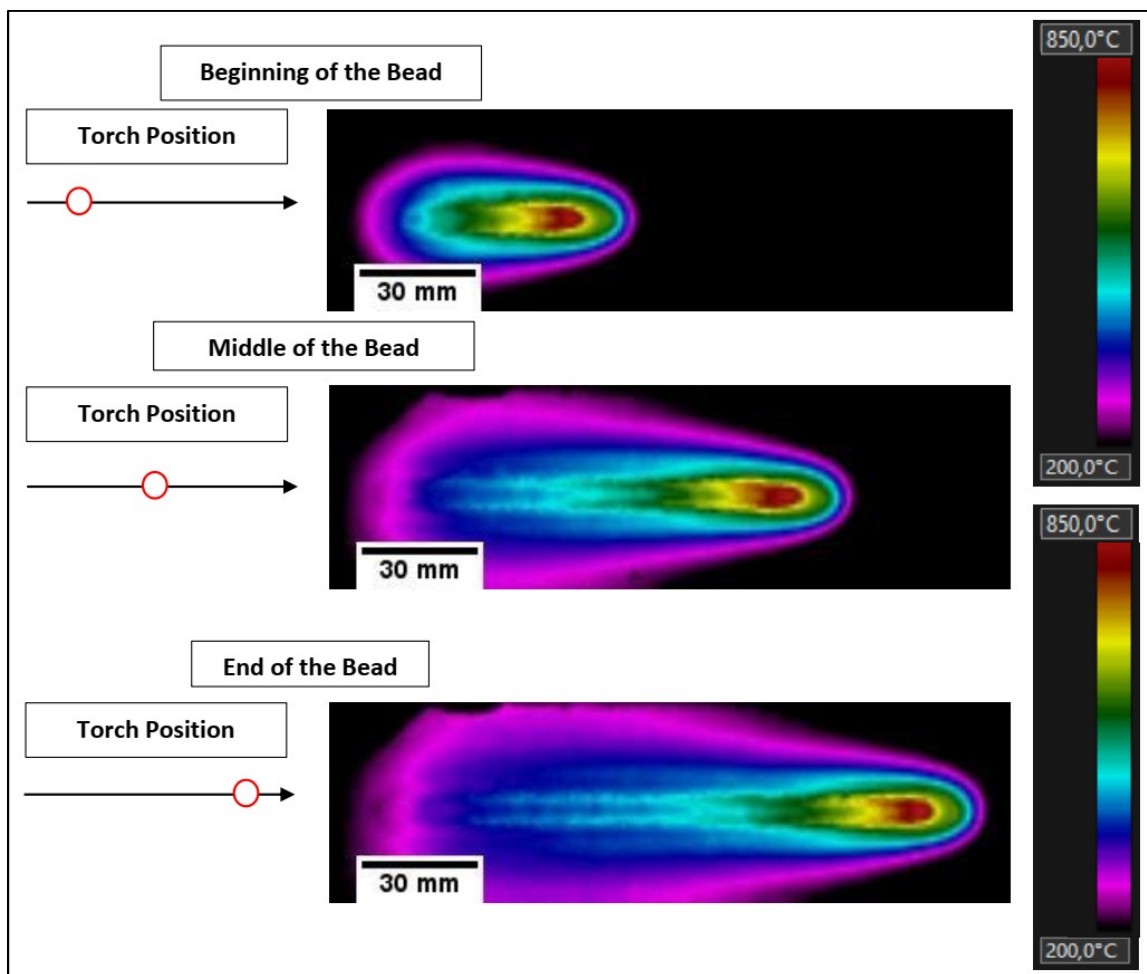

**Figure 9.** Thermal profile at the plate backside from the deposition T1L (linear) at different welding times (progressively advancing the torch position).

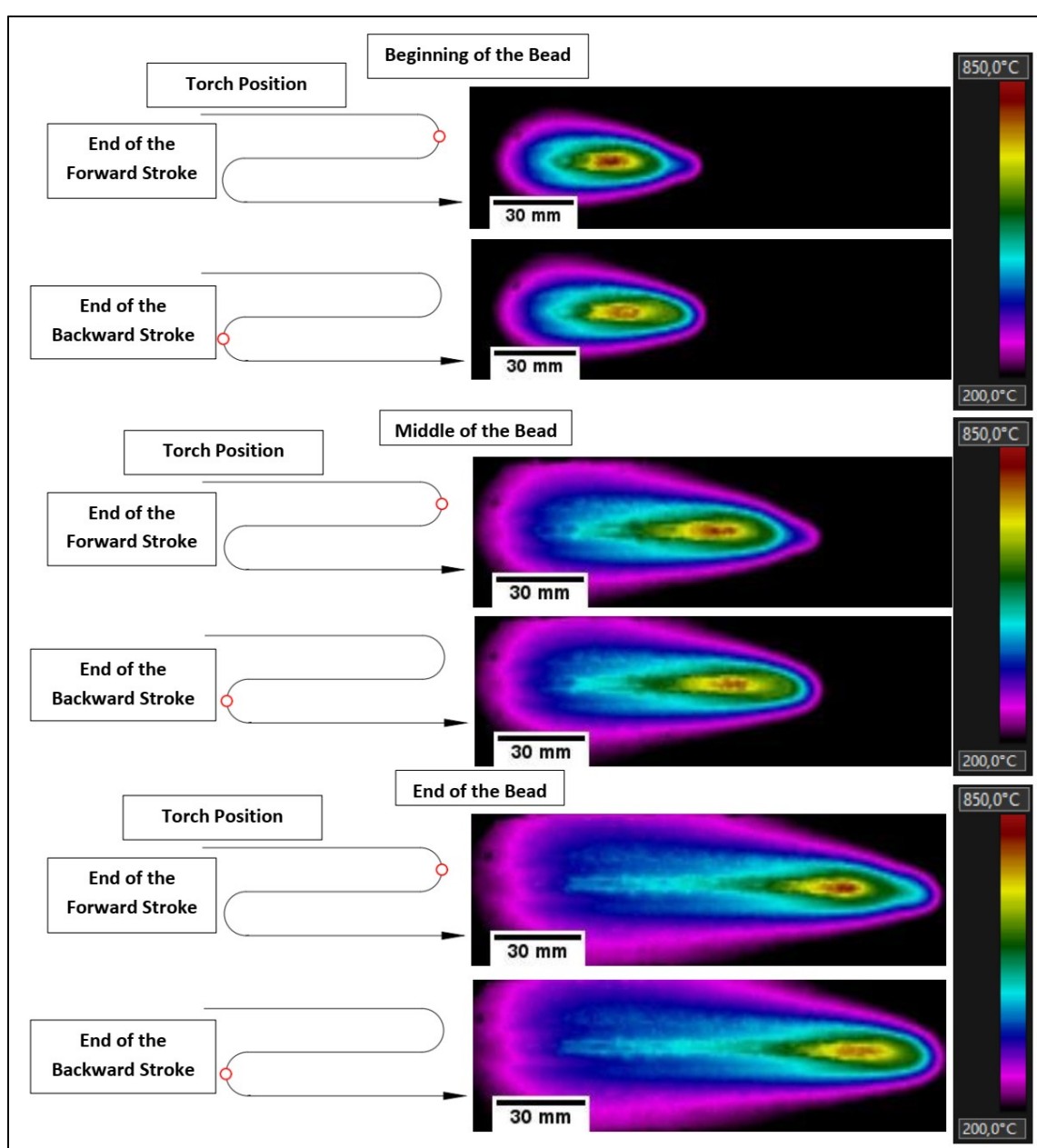

**Figure 10.** Thermal profile at the plate backside from the deposition T1SB (switchback with F = 30 mm and B = 20 mm) at different welding times (progressively advancing the torch position).

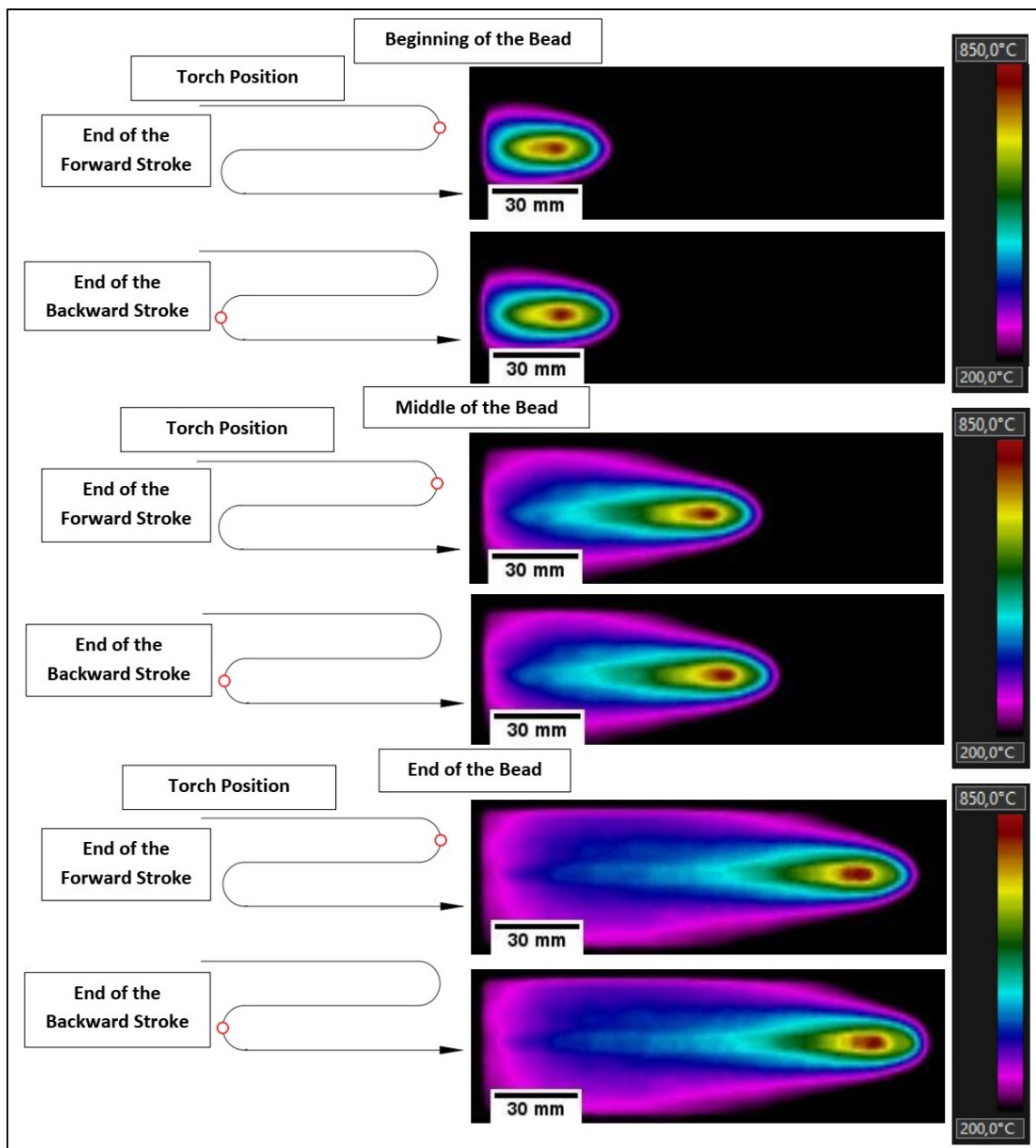

**Figure 11.** Thermal profile at the plate backside from the deposition T2SB (switchback with F = 10 mm and B = 5 mm) at different welding times (progressively advancing the torch position).

After evaluating the isotherm format of the red areas, it is clear that the pool from the linear movement was larger than in the plates in which switchback was applied (although the bead volume was the same). The larger the stroke lengths and the faster stroke speeds (T1SB in relation to T2SB), the smaller the weld pool. For a better comparison, Figure 12 presents the area values corresponding to the red isotherm as the arc progressed with the welding torch under the conditions T1L, T1SB, and T2SB, where the suffixes F and B for switchback conditions indicate the measurements at the end of the forward and backward strokes, respectively.

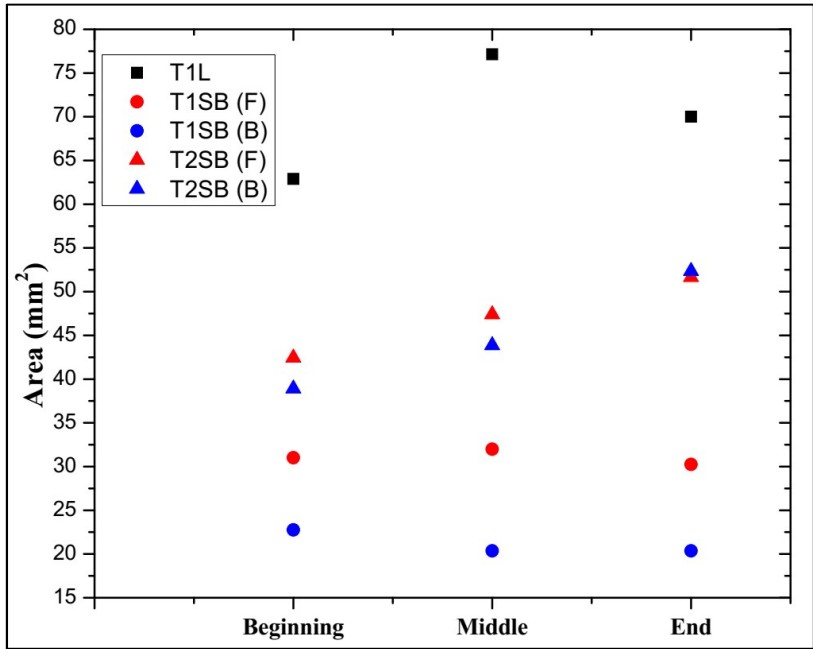

**Figure 12.** Areas of the hottest points at the plate backside, defined by the red isotherms, at different torch positions (progressively advancing): T1L (linear); T1SB (switchback with F = 30 mm and B = 20 mm); T2SB (switchback with F = 10 mm and B = 5 mm).

Another way to see the effect of lengths and speeds on the bead formation is based on the maximum temperatures within the red isotherms. Figure 13 presents the behavior of these temperatures along the time. As seen, a higher and slightly more uniform maximum temperature range was identified for the linear deposition in relation to switchback depositions. Deposits T1SB (switchback with F = 30 mm and B = 20 mm) and T2SB (switchback with F = 10 mm and B = 5 mm), in turn, had temperature variations in the form of ramps, which were associated with the switchback movements exerted by the welding torch (clearer for T1SB, lower backward stroke frequency). It should be noted that these values, although following a logical trend, are very close to each other. However, it should also be noted that these temperatures were measured on the plate underside. Thus, the heat had to cross more than half thickness of the plate to heat up that backside, distributing to a certain level before reaching the opposite face. In this way, it is envisaged that these differences would be greater if the temperature measurements were made on the top of the plates, although other sources of errors would appear. The same average temperatures, now discretized and plotted in Figure 14, make clear not only the difference between the linear and switchback conditions, but also between long (T1SB) and shorter (T2SB) strokes.

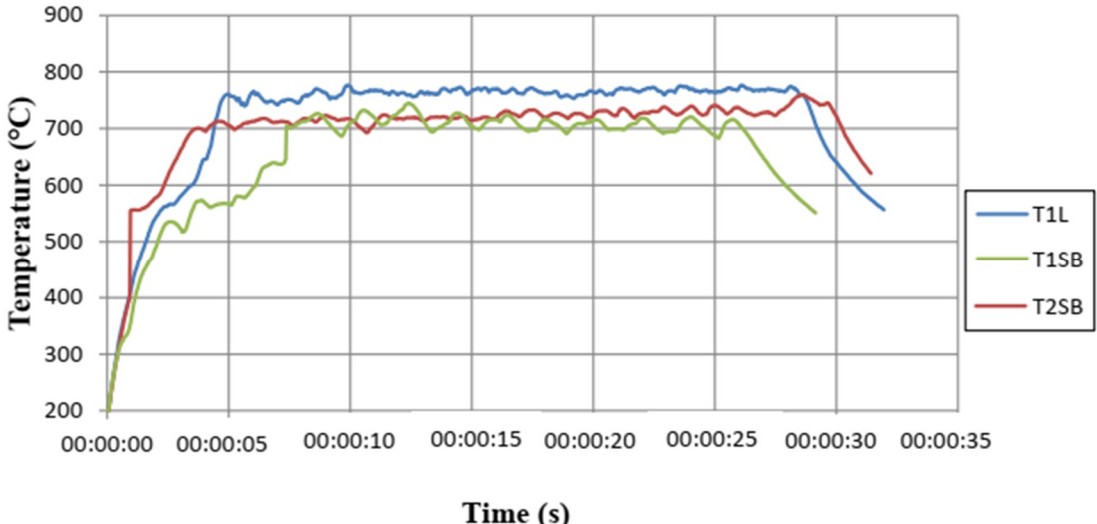

**Figure 13.** Maximum temperatures within the red isotherms, at different welding times (progressively advancing the torch position): T1L (linear); T1SB (switchback with F = 30 mm and B = 20 mm); T2SB (switchback with F = 10 mm and B = 5 mm).

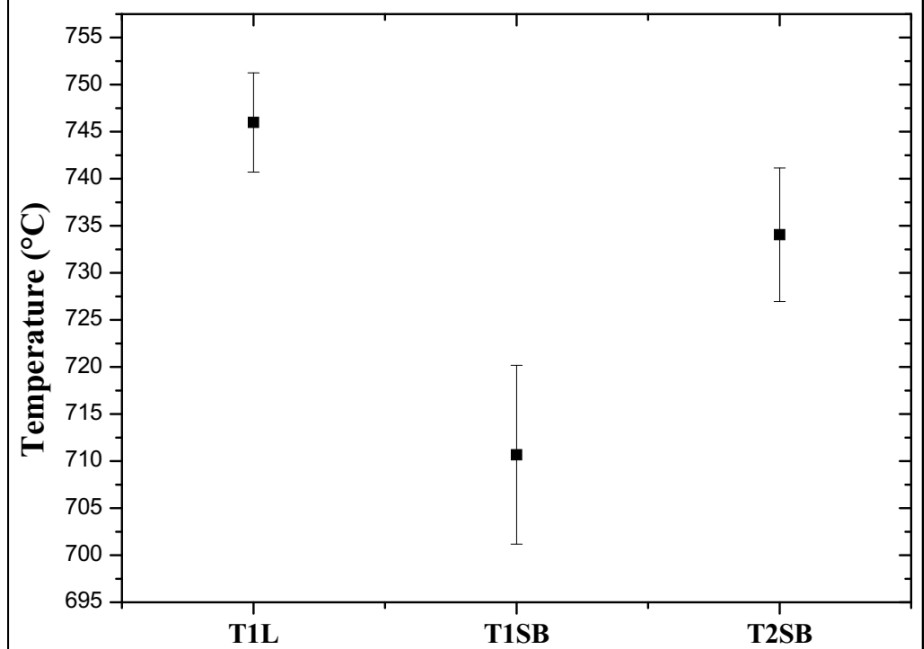

**Figure 14.** Average temperatures within the red isotherms during the depositions: T1L (linear); T1SB (switchback with F = 30 mm and B = 20 mm); T2SB (switchback with F = 10 mm and B = 5 mm).

In summary, the linear movement produced larger pools and higher maximum temperatures on the backside of the plate than when switchback was used. By comparing the switchback with lesser lengths and speeds (T2SB) to that with higher lengths and speeds (T1SB), the first one presented larger pools and higher maximum temperatures. It is important to state that these results are in agreement with the simulations carried out by Kaneko et al. [5,6], when the forward stroke speed was faster than that of the backward stroke speed. According to them, the weld pool length became shorter with switchback than a constant travel speed. Schwedersky et al. [14], also using thermography on the back of the plate, suggested that the heat tends to penetrate less into the plate when switchback is applied.

One can assume that hotter and larger weld pools would lead to slower cooling rates. Slower cooling rates at a high temperature, in turn, would keep a lower molten metal viscosity for a longer time, a fact that, along with a higher molten volume (higher gravity force), favors the pool collapse before solidification and makes it difficult to sustain it with wider gaps. This reasoning could explain the results of Section 2, in which the conditions with switchback reached wider root gaps (see Table 3). The higher temperature of the linear movement would justify the possibility of reaching full penetration with a very low gap opening. However, this reasoning by itself would not explain the variation in operational gap ranges for different switchback parameters. For instance, in terms of the switchback parameters, the condition of T2SB (Section 3) is very similar to condition 5SB (Section 2) which reaches wider root gaps, but also a gap range narrower than that with linear movement and other conditions in which $TS_F$ is smaller or equal to $TS_B$. Thus, it becomes clear that there is a dependence between the stroke lengths and the stroke speeds in the influence of the switchback parameterization.

## 4. Conclusions

The results showed that the switchback technique can be successfully used to sustain the pool in the root of the joint with wider root gaps, although the operational gap range is not necessarily larger when switchback is applied. There is a strong influence and interaction of the values of the stroke lengths and stroke speeds on the process performance.

Nevertheless, in general, the results showed that for very narrow root gaps, the use of switchback does not seem to be the best option (linear movement leads to a larger weld pool to guarantee the total penetration). For intermediary root gaps, the best switchback conditions seem to be with shorter stroke lengths and slower stroke speeds (intermediate pool size), while for larger root gaps, the best switchback conditions seem to be with longer stroke lengths and faster stroke speeds (smaller pool size, more easily sustainable).

Based on these trends, an ideal welding system for butt welds with variable root gaps can be realized, in which a sensor would measure the gap ahead of the heat source, in order to have movement and parameters adaptively changed according to the root gap value.

**Author Contributions:** Conceptualization, H.A.L.d.A. and A.S.; Methodology, H.A.L.d.A., A.S., F.R.T. and C.A.M.d.M.; Validation, H.A.L.d.A., A.S., F.R.T.; Formal analysis, F.R.T.; Investigation, H.A.L.d.A. and F.R.T.; Writing—original draft preparation, H.A.L.d.A., F.R.T. and C.A.M.d.M.; Writing—review and editing, F.R.T. and A.S.; Project administration, A.S.; Funding acquisition, A.S. and C.A.M.d.M.

**Funding:** This research was funded by CAPES, grant number 001, National Council for Scientific and Technological Development—CNPq, grant numbers 302863/2016-8 and 01.10.0723.00, PETROBRAS/CENPES and REMULT (Nucleus of Union and Coating of Materials).

**Conflicts of Interest:** The authors declare no conflict of interest. The funders had no role in the design of the study; in the collection, analyses, or interpretation of data; in the writing of the manuscript, or in the decision to publish the results.

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
