# Peer review of "The Effect of Switchback Parameters on Root Pass Formation of Butt Welds with Variable Gap"

_jmmp, doi:10.3390/jmmp3030054_

Round 1
Reviewer 1 Report
The paper may be published in its current form.
Author Response
The authors thank the reviewer for the words and encouragement.
Reviewer 2 Report
The present manuscript provided a really detailed investigation on the effect of switchback techniques on root pass formation during butt welding of steel. Welding setup and regarding reasons were presented in an organized method. I have a few suggestions for authors to make this paper even more informative and clear to readers.
Introduction:
(1) INTRODUCTION section is a little lengthy. Authors can simplify the content from line 31 to 85 and then introduce the necessities of applying switchback technique in industries.
Materials and Methods:
(1) I know section 2.1 and 2.2 in MATERIALS AND METHODS are directly correlated to section 3.1 and 3.2 in RESULTS AND DISCUSSION. But the name of section 2.1 and 2.2 should not be the same with 3.1 and 3.2, respectively since the function of 2.1 and 2.2 is to introduce experimental setup but not investigation of effect of parameters on experimental results.
Results and Discussion:
(1) Pictures of plate surface and back shown in Figures 3-8 should be divided at least by some evident lines.
In general, authors should also provide some relevant mechanical response and microstructural characterizations of welds obtained via different welding setup to display the effect of switchback techniques on weld quality.
Author Response
a) First suggestion
(1) INTRODUCTION section is a little lengthy. Authors can simplify the content from line 31 to 85 and then introduce the necessities of applying switchback technique in industries.
Answer: Thanks to the reviewer for the remark. The authors agree that the introduction is lengthy as pointed out by the acutely reviewer. The reason for this is that the main action of the switchback is the weld pool control. However, if all the other papers published on the subject (which are not in a great number, likely all covered in this paper) are closely checked, the researchers have not given attention to the fundamentals, sticking more on the application and technological aspects of the technique. Thus, the lines from 31 to 85 aimed to introduce this aspect to the reader, i.e., to present the fundamentals that eventually would support the discussion. It is important to mention that the description on how the weld pool is formed during the arc welding process stages (under the action of the arc and after, cooling from liquid to the solid state) is not found in the welding current literature, as much as the knowledge of the authors is concerned, at least with this comprehensive emphasis (as a descriptive model). The authors believe that, if their description makes sense, this aspect can raise attention of researchers interested also on the physical aspects of the phenomenon, aggregating more value to the paper (and motivating more readers and citations). Therefore, they prefer to keep the introduction as it is, at a cost of its long length.
b) Second suggestion
Materials and Methods:
(1) I know section 2.1 and 2.2 in MATERIALS AND METHODS are directly correlated to section 3.1 and 3.2 in RESULTS AND DISCUSSION. But the name of section 2.1 and 2.2 should not be the same with 3.1 and 3.2, respectively since the function of 2.1 and 2.2 is to introduce experimental setup but not investigation of effect of parameters on experimental results.
Answer: Very interesting point. Thanks for the observation. Considering that those subsections are under the umbrella of the sections 2. Materials and Methods and 3. Results and Discussion, respectively, without any introduction (to make the text shorter), the authors suggest the following changes in the subsections titles to fulfill the reviewer´s concern:
2.1. Experimental planning to evaluate the effect of stroke lengths and speeds on the operational gap range limits
3.1. Effect of stroke lengths and speeds on the operational gap range limits
2.2. Experimental planning to evaluate the effect of stroke lengths and speeds on the size of the weld pool
3.2. Effect of stroke lengths and speeds on the size of the weld pool
c) Third suggestion
Results and Discussion:
(1) Pictures of plate surface and back shown in Figures 3-8 should be divided at least by some evident lines.
Answer: Thanks for the suggestion. A while solid line was placed to split the two views. The original pictures were modified and uploaded again.
d) Fourth suggestion
In general, authors should also provide some relevant mechanical response and microstructural characterizations of welds obtained via different welding setup to display the effect of switchback techniques on weld quality.
Answer: Thanks for the suggestion. There is a paper under development in our group that deals with the effect of the switchback on microstructure. Unfortunately, especially because its experimental concept is different from the present paper, to joint both subject together would make the paper too long.
Reviewer 3 Report
This manuscript reports an experimental investigation on the operational gap ranges of conventional and switchback arc welding. The temperature was also measured to gain insights. The manuscript is well written. The following issues should be addressed during revision.
1. A comprehensive review of knowledge and literature is presented in the introduction section. The reviewer suggests the authors highlight the novelty and new findings of their research in the context of the literature, and add more discussion about the problem unsolved or less known, e.g. the combined effects of stroke time and length.
2. What are the challenges in terms of the operation of switchback arc welding? How does the findings of this research guide operation? More information about operation and more discussion about the challenges should be added.
3. In Figure 12, why the hottest areas of T1SB(F) and T1SB(B) at the middle and end are so different?
Author Response
a) First suggestion
1. A comprehensive review of knowledge and literature is presented in the introduction section. The reviewer suggests the authors highlight the novelty and new findings of their research in the context of the literature, and add more discussion about the problem unsolved or less known, e.g. the combined effects of stroke time and length.
Answer: The authors thank for the reviewer’s suggestions. As mentioned in the Introduction, despite the fact that GMAW switchback has been acknowledged for some years, there is no much information in current literature about this technique. One reason for this fact would be lack of robust information on parametrization, fact that would meet the reviewer´s concerns. It means that there are endless topics for research on this field. However, the objective of this specific work was dedicated to root gap tolerances (operational gap range) when switchback replaces the conventional linear progression welding. It would be difficult to cover other aspects in just one paper.
On the other hand, the authors would like to mention that other papers about switchback parametrization were already published by the authors, accomplishing to some extent the reviewer´s requests(the first reference was included in th text, considering the the seond one had already been cited):
Teixeira, F.R.; Mota, C.A.M.; Almeida, H.A.L.; Scotti, A.. Operational Behavior Of The Switchback Gmaw Process Using a Mechanized Rig For Arc Movement, Journal Of Materials Processing Technology, 269, 2019, Pp. 135-149, Issn 0924-0136, Doi: 10.1016/j.Jmatprotec.2019.02.014.
Almeida, H.A.L.; Mota, C.A.M.; Scotti, A. Effects of the reversion course length and torch leading angle on the bead solidification structure in GMAW welding with Switchback. Soldagem Inspeção 2012, 17(2): 123-137 (in Portuguese). DOI: 10.1590/S0104-92242012000200006.
b) Second suggestion
2. What are the challenges in terms of the operation of switchback arc welding? How does the findings of this research guide operation? More information about operation and more discussion about the challenges should be added.
Answer: The authors agrees with the reviewer´s questioning. However, to attend all of them in a detailed way would demand a greater number of pages, usually not accepted by editors. Somehow, the challenges are expressed in the Introduction section, based on the references. To guide the operational discussion, some finds are covered in a summary way in the conclusion, such as:.
- There is a strong influence and interaction of the values of the stroke lengths and stroke speeds on the process performance.
- For very narrow root gaps, the use of switchback does not seem to be the best option (linear movement leads to larger weld pool to guarantee the total penetration).
- For intermediary root gaps, the best switchback conditions seem to be with shorter stroke lengths and slower stroke speeds (intermediate pool size),
- while for larger root gaps, the best switchback conditions seem to be with longer stroke lengths and faster stroke speeds (smaller pool size, more easily sustainable).
And, concerning the gaps:
- Based on these trends, it can be idealized a welding system for butt welds with variable root gaps, in which a sensor would measure the gap ahead of the heat source, in order to have movement and parameters adaptively changed accordingly with the root gap value.
c) Third suggestion
3. In Figure 12, why the hottest areas of T1SB(F) and T1SB(B) at the middle and end are so different?
Answer: The authors thank for the suggestion from a sharp-eyed reviewer. This behaviour helps a lot the quality of the work. In fact, an error occurred during these measurements and was only noticed with this comment. The measurements of the hottest areas occurred at relatively close time range for the three depositions. However, as observed in Figure 13, the T1SB experiment takes more time to start a regime behavior than the others, requiring that its measurement at the beginning with a little more time. Thus, these measurements were redone and Figure 12 was replaced. In the new Figure 12, there are considerable differences for the hottest areas of T1SB (F) and T1SB (B) at the beginning, middle and end. This behavior can be understood as a reflection of the higher temperature variations that were observed for this condition, as shown in Figures 13 and 14.